# Molecular Screening and Genetic Identification of *Anaplasma platys* in Brown Dog Tick (*Rhipicephalus sanguineus s. l.*) Infested on Stray Dogs in Taiwan

**DOI:** 10.3390/microorganisms12091779

**Published:** 2024-08-28

**Authors:** Li-Lian Chao, Pei-Yin Ko, Chien-Ming Shih

**Affiliations:** 1M.Sc. Program in Tropical Medicine, College of Medicine, Kaohsiung Medical University, Kaohsiung 807378, Taiwan; d91632003@gmail.com; 2Graduate Institute of Pathology and Parasitology, National Defense Medical Center, Taipei 114201, Taiwan; 3Department of Medical Research, Kaohsiung Medical University Hospital, Kaohsiung 807377, Taiwan

**Keywords:** *Anaplasma platys*, genetic identity, *Rhipicephalus sanguineus*, Taiwan, tick

## Abstract

*Anaplasma platys* is a tick-borne zoonotic pathogen of canines. In this study, the presence of *A. platys* was screened for in brown dog ticks (*Rhipicephalus sanguineus s. l.*) infesting stray dogs in Taiwan to determine overall prevalence. This study represents the first instance of genetic identification of *A. platys* in brown dog ticks in Taiwan. In total, we examined 324 brown dog ticks for *A. platys* infection by nested polymerase chain reaction assay targeting the 16S ribosomal RNA gene. The general prevalence of *A. platys* infection was 3.1%, with 3.6%, 4.0%, and 2.1% in nymph, female, and male ticks, respectively. Monthly prevalence of infection was observed from May to September. Genetic relatedness was determined by comparing the sequences of the 16S rRNA gene obtained from six Taiwan strains and seventeen other strains, representing six genospecies of *Anaplasma* spp. and three outgroups (*Ehrlichia canis*, *Rickettsia rickettsia,* and *Escherichia coli*). All Taiwan specimens were shown to genetically belong to the *A. platys* group, and could be clearly discriminated from other *Anaplasma* spp. Genetic similarities revealed a 100% identity match with various *A. platys* documented in GenBank. This study highlights the epidemiological importance of geographical transmission of *A. platys* among dogs and the possible risk for human infections in Taiwan.

## 1. Introduction

*Anaplasma platys* (formerly *Ehrlichia platys*) is a Gram-negative intracellular rickettsiae that resembles a blue intraplatelet organism in a stained blood smear. It is recognized as the etiological agent for cyclic thrombocytopenia in canines [1,2,3]. This canine infection has been demonstrated around the world, distributed across various countries in North America [4,5], Central and South America [6,7,8], Europe [9,10,11,12,13], and Asia [14,15,16,17,18,19]. Human infections with *A. platys* have also been reported, based on the blood of infected persons [7,20,21,22]. The brown dog tick (*Rhipicephalus sanguineus s. l.*) has been considered the primary vector for the transmission of *A. platys* [23,24,25]. Therefore, epidemiological surveying of this zoonotic agent in brown dog ticks is essential to realize the possible risk of emerging tick-borne *A.* platys infections in the Taiwan area.

The brown dog tick is a bloodsucking arthropod that has emerged as the dominant ectoparasite infesting canines in tropical and subtropical countries [26,27]. In addition, this tick species has been identified as a major vector for various zoonotic tick-borne pathogens, including *Babesia* spp., *Ehrlichia* spp., and *Rickettsia* spp., that are transmitted among vertebrate hosts [28,29,30]. The increasing detection of *Babesia* and *Rickettsia* infections in brown dog ticks in Taiwan [31,32,33,34] and the associated medical and veterinary importance of these infections have resulted in increased research attention focusing on investigating zoonotic tick-borne pathogens in brown dog ticks in Taiwan. Although this tick species has been recognized as a primary vector for various zoonotic tick-borne pathogens, the prevalence of *A. platys* infection and the genetic identity of this pathogen have never been identified in *R. sanguineus* ticks in Taiwan.

Molecular analysis of genetic relatedness by comparing the sequence variation of individual base-pairs provides direct evidence for determining the genetic divergence between and within species of *Rickettsia* [35,36]. Indeed, previous studies according to the 16S rRNA gene of the *A. platys* strains identified from dogs have revealed that it is instructive for the phylogenetic analysis of the genetic relatedness of *Anaplasma* pathogens from different areas [10,11,12,13,14,15,16,17,18]. Therefore, phylogenetic analysis according to the genetic comparison of the 16S rRNA gene has made it feasible to determine the genetic relatedness of *Anaplasma* spp. within vector ticks.

The proposed aims of the present study are to screen for *Anaplasma* infection in brown dog ticks (*R. sanguineus s. l.*) infesting stray dogs in Taiwan, and to identify the *Anaplasma* spp. by comparing with the existing common genospecies of *Anaplasma* recorded from different geographical areas and biological origins in GenBank.

## 2. Materials and Methods

### 2.1. Collection and Identification of Tick Species

All *R. sanguineus* ticks infesting stray dogs were collected monthly from several localities in the Wanhua district of Taipei City, located in northern Taiwan. All partially engorged ticks were collected from infested dogs and subsequently stored in separate glass vials. The tick species was identified according to the pictorial keys of *R. sanguineus s. l.* [31]. The morphological structures of the *R. sanguineus s. l.* ticks were observed with a Nikon stereo-microscope (Model SMZ 1500, Tokyo, Japan) and photographed for species identification. Molecular identification by targeting the 16S mitochondrial DNA was also performed for species identification, as described previously [31].

### 2.2. Genomic DNA Extraction

In this study, each tick specimen was used for extraction of total genomic DNA. Individual tick specimens were cleaned by sonication for 3–5 min in 75% ethanol solution and then washed twice with sterile distilled water. Thereafter, the individual tick specimens were placed in an Eppendorf microtube containing 180 μL lysing buffer supplied from a DNeasy Blood & Tissue Kit (catalogue no. 69506, Qiagen, Taipei, Taiwan) and then homogenized with a TissueLyser II apparatus (catalogue no. 85300, Qiagen, Hilden, Germany). After centrifugation, the supernatant fluid was further processed by a DNeasy Blood & Tissue Kit. Thereafter, the DNA-containing fluid was collected, and the DNA concentration was quantitated with an Epoch spectrophotometer (Biotek, Winooski, VT, USA). All extracted DNA specimens were stored in a freezer (−80 °C) until further assays were due to be performed.

### 2.3. Molecular Screening for Anaplasma Infection in Collected Ticks

Extracted DNA specimens from each tick were used as a DNA template for screening of *Anaplasma* infection by PCR assay. According to the 16S rRNA gene of *Anaplasma*, two primer sets were synthesized and used for nested PCR assays. Firstly, the primary DNA product was amplified by using the initial primer set of ECC (5′-AGAACGAACGCTGGCGGCAAGC-3′) and ECB (5′-CGTATTACCGCGGCTGCTGGCA-3′). Thereafter, the species-specific primer set of EPLAT5 (5′-TTTGTCGTAGCTTGCTATGAT-3′) and EPLAT3 (5′-CTTCTGTGGGTACCGTC-3′) targeting the DNA fragment of *Anaplasma platys* was used as the secondary primer set for amplifying a DNA product approximately 359 bp in length [37,38]. Each 25 μL reaction mixture contained 2.5 μL 10× PCR buffer (Mg^2+^), 2 μL dNTP mixture (10 mM each), 1.5 μL each of the forward and reverse primers, 1 unit of Taq DNA polymerase (Takara Shuzo Co., Ltd., Tokyo, Japan), 3 μL DNA template, with the remaining volume made up with an adequate volume of ddH_2_O. In addition, the reaction mixture without the DNA template was used as a negative control. All PCR amplifications were performed in a thermocycler (Veriti, AppliedBiosystems, Taipei, Taiwan) and the initial PCR assay was performed under a pre-cycling condition of denaturation at 94 °C for 3 min, followed by 35 cycles under the conditions of denaturation at 94 °C for 1 min, annealing at 55 °C for 2 min, and extension at 72 °C for 2 min. The nested PCR assay was performed under the same pre-cycling condition and followed by 35 cycles with the same conditions of denaturation and annealing, but extension was thereafter performed at 72 °C for 90 s, and a final extension step at 72 °C for 2 min was also added.

All resulting PCR products were loaded in wells of 1.5% agarose gel and were electrophoresed in Tris-Borate-EDTA buffer. After staining with ethidium bromide, the expected DNA bands were visualized under ultraviolet light. For measuring the molecular size, a 100-bp DNA ladder (GeneRuler, Thermo Scientific, Hsinchu, Taiwan) was used as the standard marker.

### 2.4. Gene Sequencing and Phylogenetic Analysis of Genetic Relatedness

For gene sequencing, 10 μL of each selected specimen with an expected band on the agarose gel was prepared for gene sequencing. Briefly, the sequencing process was conducted with 25 cycles of nested amplification using the Big Dye Terminator Cycle Sequencing Kit (Mission Biotech Co., Ltd., Taipei, Taiwan) under a DNA Sequencer (ABI Prism 377-96, Applied Biosystems, Foster City, CA, USA), as described previously [34]. For phylogenetic analysis, the initial sequences were edited using BioEdit software (V5.3) and aligned with the CLUSTAL W software (version X) [39]. Therefore, the aligned sequences of six Taiwan strains were compared with available sequences from GenBank, including fourteen strains of *Anaplasma* spp. and three outgroup strains documented from various geographical areas and biological origins. To estimate the entire alignment, the neighbor-joining (NJ) method was performed and compared with the maximum likelihood (ML) method using the MEGA X software package (http://www.ebi.ac.uk/clustalw/, accessed on 30 July 2024) [40]. The genetic distance values were also analyzed to reveal the inter- and intra-species variations using the Kimura two-parameter model, as described previously [41]. To evaluate the reliability of the construction, all phylogenetic trees were performed with 1000 bootstrap replications [42].

### 2.5. Registration of Nucleotide Sequences of Anaplasma platys from Taiwan

The 16S rRNA gene sequences of the six Taiwan strains of *A. platys* identified from brown dog ticks (*R. sanguineus*) of Taiwan were registered and assigned the following GenBank accession numbers: 97-TP-WH-08-sd06-EN2 (OP389147), 97-TP-WH-08-sd06-M4 (OP389149), 97-TP-WH-08-sd06-PEA2 (OP392571), 97-TP-WH-08-sd06-M10 (OP392576), 97-TP-WH-08-sd06-F4 (OP392577), and 97-TP-WH-08-sd06-F2 (OP392579). In addition, the 16S rRNA gene sequences from other fourteen strains of *Anaplasma* spp. and the three outgroup strains were included for comparison (Table 1).

## 3. Results

### 3.1. Screening for Anaplasma Platys Infection in Brown Dog Ticks (R. sanguineus) of Taiwan

*A. platys* was detected in brown dog ticks of Taiwan by nested PCR assays targeting the 16S rRNA gene. In total, 3.09% (10/324) of the examined ticks were determined to be carrying *A. platys* infection. In addition, *A. platys* infection was detected at infection rates of 3.64%, 4.03%, and 2.07% in nymph, female, and male brown dog ticks, respectively (Table 2). Monthly prevalence of *A. platys* infection was observed from May to September, and the prevalence was significantly different between seasons (Table 2).

### 3.2. Genetic Relatedness of Anaplasma platys Detected in Brown Dog Ticks (R. sanguineus) of Taiwan

The sequences of the 16S rRNA gene of six *Anaplasma* strains identified from brown dog ticks in Taiwan were analyzed with the 16S rRNA gene sequences from fourteen other *Anaplasma* strains and three outgroup strains, identified from various geographical areas and biological origins and downloaded from GenBank. Results revealed that all Taiwan strains of *Anaplasma* spp. detected in brown dog ticks were identified to have 100% sequence similarity with the genospecies of *A. platys* (Table 3). Based on the genetic distance (GD) values, inter- and intra-species analysis of 16S rRNA genes revealed no genetic variance (GD = 0) among the *A. platys* strains of Taiwan, as compared with the type strains of *A. platys* (Table 3).

### 3.3. Phylogenetic Analysis of Anaplasma platys Detected in Brown Dog Ticks (R. sanguineus s. l.) of Taiwan

Based on the sequence similarity of the 16S rRNA genes, phylogenetic relationship analysis was performed to reveal the genetic relatedness among the 20 strains of *Anaplasma* and the 3 outgroup strains. Generally, the phylogenetic relationships of the *Anaplasma* strains were constructed and analyzed using both the neighbor-joining (NJ) and maximum likelihood (ML) methods. Results demonstrated seven major clades of *Anaplasma* with congruent basal topologies that could be easily discriminated by both NJ (Figure 1) and ML analyses (Figure 2). Briefly, a monophyletic clade was constituted by all *Anaplasma* strains of Taiwan, revealing a genetic affiliation with the genospecies of *A. platys* (Figure 1 and Figure 2). In addition, there was no sequence variation within the same genospecies of *A. platys* detected in *R. sanguineus s. l.* ticks of Taiwan, but there existed higher sequence divergence from other genospecies of *Anaplasma* documented from various geographical areas and biological origins (Table 3).

## 4. Discussion

In the present study, we report the initial molecular screening of *Anaplasma* infection in brown dog ticks (*R. sanguineus s. l.*) infesting stray dogs in Taiwan, and determine the genetic identity of *A. platys* detected in brown dog ticks in Taiwan. In previous reports, *Anaplasma platys* was first described as a canine pathogen that infects the host’s platelets, and was identified as the etiological agent for canine cyclic thrombocytopenia [1,2]. This canine infection is often reported subclinically, with only mild clinical signs, throughout the world [4,5,6,8,9,10,12,13,14,15,16,17,18,19]. Severe infections have mostly been found in Europe [10,12,13]. Although the *R. sanguineus* tick is recognized as the primary vector for the transmission of *A. platys*, *A. platys* DNA has also been detected in other tick species, such as *Dermacentor auratus* from Thailand [15], and *Hemaphysalia longicornis* and *Ixodes persulcatus* from Korea [16]. However, the vector competence of these tick species requires further verification. Accordingly, our results provide the first molecular confirmation of *A. platys* identified in *R. sanguineus* ticks of Taiwan, and demonstrate the first conclusive sequences of *A. platys* discovered in *R. sanguineus s. l.* ticks of Taiwan (Table 1).

Genetic relatedness among *A. platys* strains can be verified by phylogenetic analysis of the genetic similarity of the 16S rRNA gene sequence. In previous studies, sequence comparison of the 16S rRNA gene of *A. platys* strains identified from various geographical areas and biological origins has revealed the feasibility of discriminating the phylogenetic relationships of *A. platys* among other genospecies of *Anaplasma* [5,6,7,8,12,13,14,15,16,17,18,23,25]. Indeed, *Anaplasma* can be classified into several genospecies. However, the *A. platys* species is mainly identified in canine hosts [28,30]. In the present study, genetic analysis based on phylogenetic analysis of the 16S rRNA gene sequences of *Anaplasma* strains detected in *R. sanguineus* ticks of Taiwan revealed a 100% sequence similarity with the genospecies of *A. platys* documented in GenBank (Figure 1 and Figure 2, Table 3). The *A. platys* strains of Taiwan are mainly affiliated with the *A. platys* strains identified from Thailand, India, China, and Mexico (Table 3). The genetic discrimination between the *A. platys* strains in *R. sanguineus* ticks of Taiwan and other genospecies of *Anaplasma* identified from various geographic areas and biological origins is strongly supported by phylogenetic trees constructed using either NJ or ML methods (Figure 1 and Figure 2). Therefore, this study revealed a monophyletic group of Taiwan strains, verified genetically as *A. platys*, detected in brown dog ticks (*R. sanguineus s. l.*) in Taiwan.

The seasonal variation of *A. platys* infection in brown dog ticks in Taiwan remains unclear. In this study, monthly prevalence of *A. platys* infection was observed from May to September in brown dog ticks, and the highest rate of infection was detected during the summer season (June to August) (Table 2). This result is consistent with the seasonal abundance of the tick population surveyed during the summer season, as described in our previous report [43]. Indeed, the hot climate of the summer season may enhance the searching activity of brown dog ticks for feeding opportunities on stray dogs in their natural environment. In addition, a previous study also indicated a higher prevalence of *A. platys* infection in dogs observed in a heavily tick infested kennel [19]. Thus, the abundance of tick population is highly associated with the prevalence of *A. platys* infection in either ticks or dogs.

The biological mechanisms for the natural maintenance of *A. platys* by *R. sanguineus* ticks are controversial. In a previous study, the *R. sanguineus* ticks failed to acquire detectable levels of *A. platys* from laboratory-infected dogs [44]. However, it has also been found that *A. platys* infection can be maintained transstadially from naturally infected nymphs into the adult stage of *R. sanguineus* [45]. Although it is generally accepted that transovarial transmission (TOT) by vector ticks is inefficient, persistent TOT infection of *A. platys* over multiple generations has been described in a laboratory colony of *R. sanguineus* ticks [46]. In addition, the horizontal transmission of *A. platys* may also occur between co-feeding ticks. Indeed, uninfected ticks can acquire the *A. platys* infection through feeding on blood from infected ticks parasitized on the same dog [46]. These investigations may emphasize the possible role of *R. sanguineus* ticks as vector ticks and reservoir hosts for the transmission of *A. platys* in nature.

## 5. Conclusions

This investigation describes the first molecular screening of *A. platys* detected in brown dog ticks (*R. sanguineus s. l.*) infesting stray dogs of Taiwan, and represents the first verification of the genetic identity of this pathogen in brown dog ticks in Taiwan. Due to the close contact between dogs and humans, further studies focused on the vectorial capacity of *R. sanguineus* ticks would be beneficial to realize the possible risk of human infections in Taiwan.

## Figures and Tables

**Figure 1 microorganisms-12-01779-f001:**
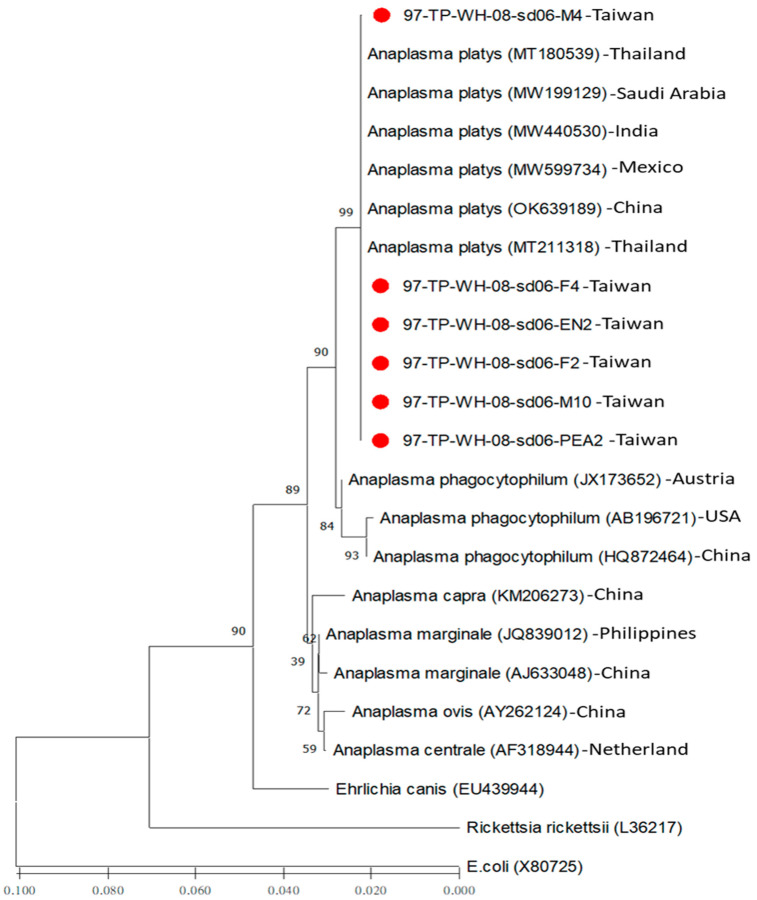
Phylogenetic analysis based on the 16S rRNA gene. The aligned sequences of 6 Taiwan strains (indicated as ●) identified in *R. sanguineus* ticks of Taiwan were compared with available sequences from GenBank, including 14 strains of *Anaplasma* spp. and 3 outgroup strains identified from different biological and geographical origins. The constructed tree was analyzed using the neighbor-joining (NJ) method using 1000 bootstrap replicates. Numbers at the nodes indicate the percentages of reliability of the tree. Branch length is drawn proportional to the estimated sequence divergence.

**Figure 2 microorganisms-12-01779-f002:**
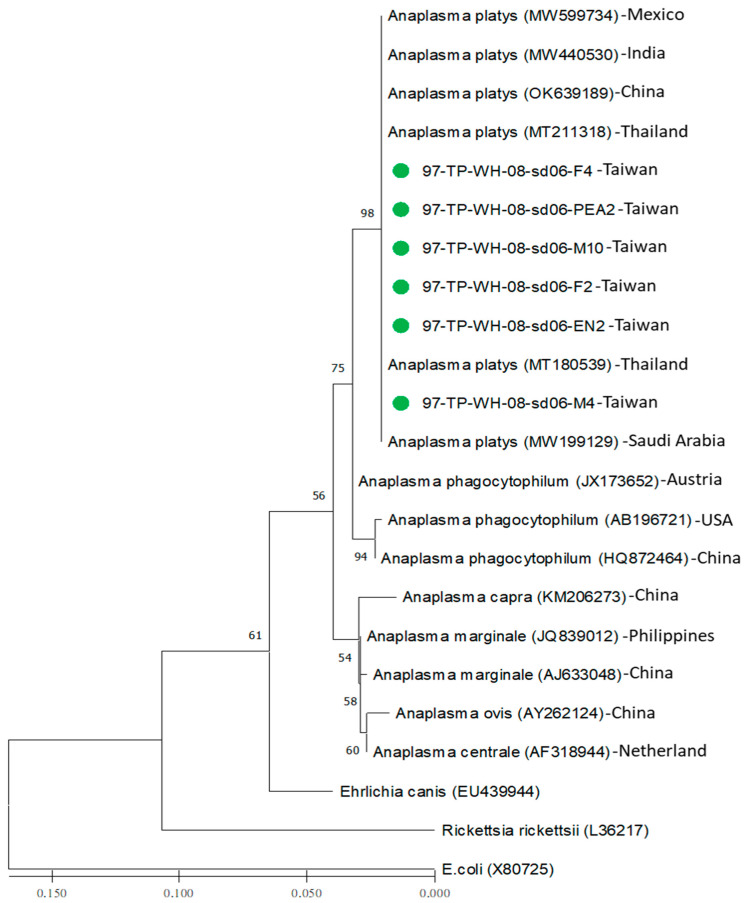
Phylogenetic analysis based on the 16S rRNA gene. The aligned sequences of 6 Taiwan strains (indicated as ●) identified in *R. sanguineus* ticks of Taiwan were compared with available sequences from GenBank, including 14 strains of *Anaplasma* spp. and 3 outgroup strains identified from different biological and geographical origins. The constructed tree was analyzed using the maximum likelihood (ML) method using 1000 bootstrap replicates. Numbers at the nodes indicate the percentages of reliability of the tree. Branch length is drawn proportional to the estimated sequence divergence.

**Table 1 microorganisms-12-01779-t001:** Phylogenetic analysis of *Anaplasma* strains of Taiwan with other documented strains from GenBank.

Strain	Origin of Bacterial Strain		16S Gene Accession Number ^a^
	Biological	Geographic	
Taiwan strain			
97-TP-WH-08-sd06-EN2	*Rhipicephalus sanguineus s. l.*	Taiwan	OP389147
97-TP-WH-08-sd06-M4	*Rhipicephalus sanguineus s. l.*	Taiwan	OP389149
97-TP-WH-08-sd06-PEA2	*Rhipicephalus sanguineus s. l.*	Taiwan	OP392571
97-TP-WH-08-sd06-M10	*Rhipicephalus sanguineus s. l.*	Taiwan	OP392576
97-TP-WH-08-sd06-F4	*Rhipicephalus sanguineus s. l.*	Taiwan	OP392577
97-TP-WH-08-sd06-F2	*Rhipicephalus sanguineus s. l.*	Taiwan	OP392579
*Anaplasma platys*	Dog blood	Thailand	MT180539
*Anaplasma platys*	Dog blood	Saudi Arabia	MW199129
*Anaplasma platys*	Dog blood	India	MW440530
*Anaplasma platys*	*Rhipicephalus sanguineus*	Mexico	MW599734
*Anaplasma platys*	Tick	China	OK639189
*Anaplasma platys*	Dog blood	Thailand	MT211318
*Anaplasma phagocytophilum*	Dog blood	Austria	JX173652
*Anaplasma phagocytophilum*	Tick	USA	AB196721
*Anaplasma phagocytophilum*	Goat	China	HQ872464
*Anaplasma capra*	Homo sapiens	China	KM206273
*Anaplasma marginale*	*Boophilus microplus*	Philippines	JQ839012
*Anaplasma marginale*	Cattle	China	AJ633048
*Anaplasma ovis*	Sheep blood	China	AY262124
*Anaplasma centrale*	Vaccine strain	Netherland	AF318944
*Ehrlichia canis*	Dog blood	Italy	EU439944
*Rickettsia rickettsia*	Unknown	France	L36217
*Escherichia coli*	Neotype strain	France	X80725

**^a^** Bold accession numbers of GenBank were submitted by this study.

**Table 2 microorganisms-12-01779-t002:** Molecular detection of *Anaplasma platys* in *Rhipicephalus sanguineus s. l.* ticks infesting stray dogs of Taiwan by nested-PCR assay targeting the 16S ribosomal RNA gene.

Collection Month	*Anaplasma platys* Detected in Various Life-Stage of Tick			Total
	Nymph (P/E) ^a^	Female (P/E) ^a^	Male (P/E) ^a^	No. Positive/ No. Examined (%)
January	0/2	0/4	0/4	0/10 (0.00)
February	0/2	0/5	0/5	0/12 (0.00)
March	0/2	0/10	0/15	0/27 (0.00)
April	0/3	0/10	0/12	0/25 (0.00)
May	0/2	1/12	0/10	1/24 (4.17)
June	1/5	1/12	0/14	2/31 (6.45)
July	1/8	1/12	1/16	3/36 (8.33)
August	0/14	2/23	1/33	3/70 (4.29)
September	0/6	0/15	1/16	1/37 (2.70)
October	0/4	0/7	0/5	0/16 (0.00)
November	0/4	0/6	0/5	0/15 (0.00)
December	0/3	0/8	0/10	0/21 (0.00)
Total (%)	2/55 (3.64)	5/124 (4.03)	3/145 (2.07)	10/324 (3.09)

**^a^** P/E = No. positive/no. examined.

**Table 3 microorganisms-12-01779-t003:** Intra- and inter-group analysis of genetic distance values ^a^ based on the 16S rRNA gene sequences between the *Anaplasma* strains of Taiwan and other *Anaplasma, Ehrlichia*, *Rickettsia,* and *E. coli* strains documented in GenBank.

Bacterial Strains ^b^	1	2	3	4	5	6	7	8	9	10	11	12	13	14	15	16	17
1. 97-TP-WH-08-sd06-M4 (Taiwan)	–																
2. 97-TP-WH-08-sd06-EN2 (Taiwan)	0.00	–															
3. 97-TP-WH-08-sd06-F2 (Taiwan)	0.00	0.00	–														
4. 97-TP-WH-08-sd06-M10 (Taiwan)	0.00	0.00	0.00	–													
5. 97-TP-WH-08-sd06-PRA2 (Taiwan)	0.00	0.00	0.00	0.00	–												
6. 98-TP-WH-08-sd06-F4 (Taiwan)	0.00	0.00	0.00	0.00	0.00	–											
7. *Anaplasma platys* (MT180539)	0.00	0.00	0.00	0.00	0.00	0.00	–										
8. *Ana. platys* (OK639189)	0.00	0.00	0.00	0.00	0.00	0.00	0.00	–									
9. *Ana. platys* (MW599734)	0.00	0.00	0.00	0.00	0.00	0.00	0.00	0.00	–								
10. *Ana. platys* (MW440530)	0.00	0.00	0.00	0.00	0.00	0.00	0.00	0.00	0.00	–							
11. *Ana. phagocytophilum* (AB1967212)	0.02	0.02	0.02	0.02	0.02	0.02	0.02	0.02	0.02	0.02	–						
12. *Ana. marginale* (JQ839012)	0.02	0.02	0.02	0.02	0.02	0.02	0.02	0.02	0.02	0.02	0.02	–					
13. *Ana. ovis* (AY262124)	0.03	0.03	0.03	0.03	0.03	0.03	0.03	0.03	0.03	0.03	0.03	0.04	–				
14. *Ana. capra* (KM206273)	0.03	0.03	0.03	0.03	0.03	0.03	0.03	0.03	0.03	0.03	0.03	0.05	0.02	–			
15. *Ehrlichia canis* (EU439944)	0.06	0.06	0.06	0.06	0.06	0.06	0.06	0.06	0.06	0.06	0.06	0.07	0.06	0.07	–		
16. *Rickettsia rickettsii* (L36217)	0.20	0.19	0.19	0.19	0.19	0.19	0.20	0.19	0.19	0.19	0.19	0.19	0.16	0.18	0.17	–	
17. *Escherichia coli* (X80725)	0.29	0.29	0.29	0.29	0.29	0.29	0.29	0.29	0.29	0.29	0.29	0.29	0.28	0.27	0.28	0.33	–

^a^ The pairwise distance calculation was performed using the Kimura 2-parameter method, as implemented in MEGA X (Kumar et al., 2018 [40]). ^b^ Strains 7–14, 15, 16, and 17 indicate the *Anaplasma*, *Ehrlichia*, *Rickettsia*, and *E. coli* strains documented in GenBank, respectively.

## Data Availability

The original contributions presented in the study are included in the article, further inquiries can be directed to the corresponding author.

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
