# Peer review of "Molecular Screening and Genetic Identification of Anaplasma platys in Brown Dog Tick (Rhipicephalus sanguineus s. l.) Infested on Stray Dogs in Taiwan"

_microorganisms, 2024, doi:10.3390/microorganisms12091779_

Round 1
Reviewer 1 Report
Comments and Suggestions for Authors
The manuscript entitled “Molecular Screening and Genetic Identification of Anaplasma platys in Brown Dog Tick (Rhipicephalus sanguineus) Infested on Stray Dogs in Taiwan” constitutes the first study in the country confirming the presence of A. platys in R. sanguineus ticks. The manuscript has an interesting discussion covering the main points. However, I have some questions and recommendations as follows:
Line 105: change “10-μl” to 10 μl
Please, check the word “outgroup” across the manuscript. It is written as “ourgroup” in several sentences (including figures).
Line 117: Why did the authors select the Kimura two-parameter model in the ML phylogenetic reconstruction? Did you test the best evolution model for your dataset?
Lines 134-135: How many samples were positive for Anaplasma in the first PCR before using the nested PCR? Could the ticks have other species of Anaplasma than A. platys?
The authors did not include ethical considerations or the required permissions in this study. Please, include it.
There is no information about the sequence length of the 16S rRNA gene sequence used for comparisons and phylogenetic analysis. Did you obtain the full 16S rRNA gene sequence, or is it a partial sequence? More precise data about this step should be provided for readers.
Table 3: I suggest using just two decimal places instead of three. The values inside are not possible to read accurately because they look together.
Lines 164-166: The sentence “The repeatability of the specimen clusters represented in phylogenetic trees was evaluated by bootstrap analysis” is part of the methodology and not the results.
Neighbour-Joining is not a phylogenetic method but a phenetic one. It establishes relationships between sequences based solely on their genetic distance, without considering an evolutionary model. Therefore, the NJ tree (Fig. 1) is not necessary for the manuscript.
Line 196: “A. platys” should be modified to Anaplasma platys since it starts the sentence.
Discussion
The 16S rRNA gene sequence revealed a 100% identity with the other ones from the same species across the globe. Is there no variation in this species, or does the molecular marker has not enough resolution to deep into the genetic variability of this species?
Comments on the Quality of English LanguageThough I am not an English native, an overall English revision is required to improve it across the manuscript.
Author Response
Responses to the Reviewer #1:
Line 105: change “10-μl” to 10 μl
Reply: As suggested, we have corrected the word (Line 108)
Please, check the word “outgroup” across the manuscript. It is written as “ourgroup” in several sentences (including figures).
Reply: As suggested, we have corrected the word (Lines 115, 129, 180, 189).
Line 117: Why did the authors select the Kimura two-parameter model in the ML phylogenetic reconstruction? Did you test the best evolution model for your dataset?
Reply: As our previous works and other studies, this model is the most popular and reliable model for comparing the genetic distance of intra- and inter-species variations used in the ML and NJ phylogenetic analysis.
Lines 134-135: How many samples were positive for Anaplasma in the first PCR before using the nested PCR? Could the ticks have other species of Anaplasma than A. platys?
Reply: There are 10 samples were positive for Anaplasma in our first PCR assay. Our nested PCR using the species-specific primer sets (EPLAT5/EPLAT3) targeting the DNA fragment of Anaplasma platys. Thus, there is only A. platys detected in our experiments.
The authors did not include ethical considerations or the required permissions in this study. Please, include it.
Reply: We already include the ethical statements in our manuscript (Page 9, Lines 262-264).
There is no information about the sequence length of the 16S rRNA gene sequence used for comparisons and phylogenetic analysis. Did you obtain the full 16S rRNA gene sequence, or is it a partial sequence? More precise data about this step should be provided for readers.
Reply: We already indicate the DNA fragments of 16S rRNA with 359-bp (partial sequence) of A. platys were used for our comparisons and phylogenetic analysis (Lines 89-90).
Table 3: I suggest using just two decimal places instead of three. The values inside are not possible to read accurately because they look together.
Reply: As suggested, We have modified the Table 3 using just two decimal and make the values inside are easily to read (Table 3, Lines 157-158).
Lines 164-166: The sentence “The repeatability of the specimen clusters represented in phylogenetic trees was evaluated by bootstrap analysis” is part of the methodology and not the results.
Reply: As suggested, We have deleted the sentence for clarity (Line 167).
Neighbour-Joining is not a phylogenetic method but a phenetic one. It establishes relationships between sequences based solely on their genetic distance, without considering an evolutionary model. Therefore, the NJ tree (Fig. 1) is not necessary for the manuscript.
Reply: As our previous reports and other studies (Exp. Appl. Acarol. 2017, 166, 356-362; 2021, 85, 291-304; TTBD, 2018, 9, 266-269; Parasites & Vectors 2021, 14, 191; 2022, 15, 138), we believe that NJ tree is well recognized method for phylogenetic analysis of genetic relationships of tick-borne pathogens and we prefer to keep it for verification (Figure 1, Lines 175-177).
Line 196: “A. platys” should be modified to Anaplasma platys since it starts the sentence.
Reply: As suggested, we have amended it (Line 198).
Discussion
The 16S rRNA gene sequence revealed a 100% identity with the other ones from the same species across the globe. Is there no variation in this species, or does the molecular marker has not enough resolution to deep into the genetic variability of this species?
Reply: Our comparison based on the partial sequence (359-bp) of 16S rRNA gene of Anaplasma spp. indicates a 100% identity. However, further study using the full-length sequence may display some bp variation of the same Anaplasma spp. from different geographical origins.
Reviewer 2 Report
Comments and Suggestions for Authors
Some parts of the text are confusing – the manuscript should be reviewed for English, preferably by a native speaker of the language
Title – rewrite to read as: Molecular Screening and Genetic Identification of Anaplasma
platys in Rhipicephalus sanguineus s.l. from Stray Dogs in Taiwan
Remark 1 – the authors should define ion the title whether R. sanguineus refers to sensu lato (s.l.) or sensu stricto (s.s.). This detail should also be addressed in the abstract and main text.
Abstract – rewrite to read as: Anaplasma platys is a tick-borne pathogen of canines, with zoonotic potential.
Line 11 – in brown dog TICKS
Lines 14, 15, etc. – do not use “rate” for percentages – use prevalence or proportion; use one decimal only (e.g. 3.1% instead of 3.09%)
Line 14 – adapt as: The general A. platys infection prevalence was 3.1%, with 3.6%, 4.03% and 2.07% in nymph, female and male ticks, respectively.
Line 18 – Anaplasma or Anaplasma spp.?
The same question as above for Ehrlichia and Rickettsia
Line 19 – replace “with the Anaplasma platys” with “with A. platys”
Line 20 – replace Anaplasma species with Anaplasma spp.; replace homology with identity
Line 21 – replace significance with importance
Keywords – display alphabetically
Introduction – first sentence: too long – please revise
Remark 2 – address the question of Rhipicephalus sanguineus being sensu lato (s.l.) or sensu stricto (s.s.) – the same in the Materials and Methods section
Line 37 – Anaplasma, A. platys or Anaplasma spp.?
Line 41 – please mention the species of Babesia, Ehrlichia and Rickettsia – or insert “spp.” for each genus – change accordingly throughout the manuscript
Line 55 – Anaplasma spp. – change accordingly throughout the manuscript
Line 71 – Eppendorf
Line 76 – were stored
Line 120 – avoid using abbreviations (e.g. A. platys) in subheadings and titles of tables and figures
Table 1 – present Latin names in italics
Table 2 – were differences between seasons statistically significant? Please mention in the text
Table 3 – separation between numbers is not clear – space seems not to be enough
Figures 1 and 2 – present Latin names in italics
Line 199 – correct: “asymptomatic symptoms” – adapt as: often subclinically or with with mild clinical signs throughout the world
Comments on the Quality of English Language
Moderate editing of English language required.
Author Response
Responses to the Reviewer #2:
Title – rewrite to read as: Molecular Screening and Genetic Identification of Anaplasma platys in Rhipicephalus sanguineus s.l. from Stray Dogs in Taiwan
Reply: As suggested, we have amended it (Line 2).
Remark 1 – the authors should define the title whether R. sanguineus refers to sensu lato (s.l.) or sensu stricto (s.s.). This detail should also be addressed in the abstract and main text.
Reply: As suggested, we have amended it to sensu lato (s.l.) (Lines 11, 34, 57, 65-66, 132-133, 142, 162, 173, 196, 208, 225, 250).
Abstract – rewrite to read as: Anaplasma platys is a tick-borne pathogen of canines, with zoonotic potential.
Reply: As suggested, we have modified the sentence (Line 10).
Line 11 – in brown dog TICKS
Reply: As suggested, we have changed the word (Line 11).
Lines 14, 15, etc. – do not use “rate” for percentages – use prevalence or proportion; use one decimal only (e.g. 3.1% instead of 3.09%)
Reply: As suggested, we have modified the words using prevalence and one decimal only (Line 14).
Line 14 – adapt as: The general A. platys infection prevalence was 3.1%, with 3.6%, 4.03% and 2.07% in nymph, female and male ticks, respectively.
Reply: As suggested, we have done it (Line 14).
Line 18 – Anaplasma or Anaplasma spp.?
Reply: As suggested, we have changed the word (Line 17).
The same question as above for Ehrlichia and Rickettsia
Reply: As suggested, we have modified the words (Line 18).
Line 19 – replace “with the Anaplasma platys” with “with A. platys”
Reply: As suggested, we have modified the word (Line 19).
Line 20 – replace Anaplasma species with Anaplasma spp.; replace homology with identity
Reply: As suggested, we have modified the words (Lines 19-20).
Line 21 – replace significance with importance
Reply: As suggested, we have modified the word (Line 21).
Keywords – display alphabetically
Reply: As suggested, we have done it (Line 24).
Introduction – first sentence: too long – please revise
Reply: As suggested, we have modified the sentence (Line 29-30).
Remark 2 – address the question of Rhipicephalus sanguineus being sensu lato (s.l.) or sensu stricto (s.s.) – the same in the Materials and Methods section
Reply: As suggested, we have done it (Lines (Lines 11, 34, 57, 65-66, 132-133).
Line 37 – Anaplasma, A. platys or Anaplasma spp.?
Reply: As suggested, we have done it (Line 37).
Line 41 – please mention the species of Babesia, Ehrlichia and Rickettsia – or insert “spp.” for each genus – change accordingly throughout the manuscript
Reply: As suggested, we have done it (Line 41).
Line 55 – Anaplasma spp. – change accordingly throughout the manuscript
Reply: As suggested, we have done it (Lines 55, 58, 151).
Line 71 – Eppendorf
Reply: As suggested, we have done it (Line 74).
Line 76 – were stored
Reply: As suggested, we have done it (Line 79).
Line 120 – avoid using abbreviations (e.g. A. platys) in subheadings and titles of tables and figures
Reply: As suggested, we have done it (Lines 123, 135, 145, 162).
Table 1 – present Latin names in italics
Reply: As suggested, we have done it (Lines 132-133).
Table 2 – were differences between seasons statistically significant? Please mention in the text
Reply: As suggested, we have done it (Line 141).
Table 3 – separation between numbers is not clear – space seems not to be enough
Reply: As suggested, we have modified the Table 3 with the numbers of two decimal and makes the space more clear to read (Table 3, Lines 157-8).
Figures 1 and 2 – present Latin names in italics
Reply: the figures 1 and 2 are produced by the analytical software and the letter within the figures can not changed. However, all Latin names in italics are expressed in Table 1 (Lines 132-133).
Line 199 – correct: “asymptomatic symptoms” – adapt as: often subclinically or with with mild clinical signs throughout the world
Reply: As suggested, we have done it (Line 200).
Reviewer 3 Report
Comments and Suggestions for Authors
Dear Authors,
The work aims to monitor pathogens (A. platys) in ticks from Taiwan. However, there are two major flaws that need to be reviewed: 1) The correct identification of ticks. It is already known today that R. sanguineus is a complex of species, and this was not even considered in the present manuscript; including old identification references that do not match the current taxonomy. 2) How can one be sure that DNA was actually extracted from ticks and not from hosts? This is a basic thing in molecular analysis; at the very least, some endogenous control of the reaction needs to be done so that a more appropriate analysis can be performed.

no comments
Author Response
Responses to the Reviewer #3:
The work aims to monitor pathogens (A. platys) in ticks from Taiwan. However, there are two major flaws that need to be reviewed: 1) The correct identification of ticks. It is already known today that R. sanguineus is a complex of species, and this was not even considered in the present manuscript; including old identification references that do not match the current taxonomy. 2) How can one be sure that DNA was actually extracted from ticks and not from hosts? This is a basic thing in molecular analysis; at the very least, some endogenous control of the reaction needs to be done so that a more appropriate analysis can be performed.
Reply: 1). To comply with the reviewer’s concern, we have modified the tick species as R. sanguineus s. l. throughout the manuscript.
2). Our DNA extraction is performed for the total genomic DNA that includes tick and host DNA. However, our nested PCR assays using the specific primer sets targeting the 16S rRNA fragment of Anaplasma platys that will amplify very specifically with the DNA materials within the extraction fluids. This molecular assay system is highly recognized by the scientific researchers working on the tick-borne pathogens in either ticks or hosts.
Reply: 1). Title (the tick species): we already added as R. sanguineus s. l. (Line 2).
2). Abstract (the tick species): we have done as R. sanguineus s. l. (Line 11).
3). Use a better word for “affiliated”: we have done it by using “belong to” (Line 19).
4) All collected ticks are R. sanguineus s. l., but not other tick species.
Reviewer 4 Report
Comments and Suggestions for Authors
The paper presents interesting points and contains some useful information for future epidemiological studies. However, the paper can be improved if a number of points (see below) are satisfactorily addressed before being accepted for its publication in Microorganisms.
The authors reported that the ticks were collected from dogs. There is no information at what stage of feeding individual ticks were. This is important because on the host, you can find ticks that have just attached and have not started feeding, partially engaged ticks, and fully engaged ticks, ready to leave the host. The group of ticks not yet feeding should be treated separately. The parasites found in them come from the host on which they fed in the previous developmental stage, or they were acquired through transovarial transmission. In ticks that are already feeding, the parasites could also have been acquired in this way, but also come from the host from which they were collected.
Genomic DNA extraction - there is no information on how the ticks were cleaned (liquid composition, rinsing time), what the dissolution procedure was (lysing fluid composition, lysis time)
Results: In diagrams 1 and 2, next to the names of Anaplasma strains, the organism from which it was isolated and the geographical location should be provided. This makes it easier for the reader to analyze the graph on their own, and to draw further conclusions than just that the strains are similar. These data are usually available in GenBank. If the author of a sequence belonging to a parasitic organism does not specify what host he received it from and where it came from, such a sequence is useless and should not be taken into account in the analysis.
Table 2 shows differences in Anaplasma prevalence in ticks in individual months. It is very interesting that infected ticks were found only in the months May–September. Why do the authors omit this data in the discussion? How will these differences be explained?
References - In the authors' names, diacritical marks appropriate to their languages ​​have been omitted. The correct spelling of names must be used. Example – 5, 7, 18…
Detail notices:
18 “all Taiwan strains” - the text shows the existence of one strain, why do the authors use the plural here? If they believe there is more than one strain, this should be justified.
29 “a blue intraplatelet organism appeared in a stained blood film” -not in stained film, but in plates.
38-39 “ dominant ectoparasite infesting canines around the world” – it is not true. R. sanguineus is the dominant parasite in the warm climate zone, limits of its distribution range are thought to be from latitude 20°N to below 30°S. In northern parts of Eurasia and Northern America it is absent.
87 “10X” – what does it mean? Is it 10×? The X (0078 Unicode, or 120 ASCII) sign means in the scientific record the unknown value. I suggest to replace it with the multiplication sign × (00D7 Unicode, or 215 ASCII); in computer text editors, it is no problem.
196 “A. platys was first described” - this claim is not allowed in this place of the text! The authors of the manuscript were not the first to describe A. platys! This is the first identify of Anaplasma infection in R. sanguineus ticks infestig stray dogs of Taiwan.
Author Response
Responses to the Reviewer #4:
The authors reported that the ticks were collected from dogs. There is no information
at what stage of feeding individual ticks were. This is important because on the host,
you can find ticks that have just attached and have not started feeding, partially
engaged ticks, and fully engaged ticks, ready to leave the host. The group of ticks not
yet feeding should be treated separately.
Reply: As suggested, we have added one sentence to describe that all collected ticks
were partially engorged and removed from infested stray dogs (Lines 63-4).
Genomic DNA extraction - there is no information on how the ticks were cleaned
(liquid composition, rinsing time), what the dissolution procedure was (lysing fluid
composition, lysis time).
Reply: As suggested, we have added one sentence in Materials and Methods to
describe the detailed procedures (Lines 72-75).
Results: In diagrams 1 and 2, next to the names of Anaplasma strains, the organism
from which it was isolated and the geographical location should be provided. This
makes it easier for the reader to analyze the graph on their own, and to draw further
conclusions than just that the strains are similar. These data are usually available in
GenBank. If the author of a sequence belonging to a parasitic organism does not
specify what host he received it from and where it came from, such a sequence is
useless and should not be taken into account in the analysis.
Reply: Although we already described the biological and geographical origins of
Anaplasma strains in Table 1 of our manuscript, we have added the isolation source of different countries of each Anaplasma strains in our revised Figures 1 and 2 (Table 1, Lines 175-176, 184-185).
Table 2 shows differences in Anaplasma prevalence in ticks in individual months. It is very interesting that infected ticks were found only in the months May–September. Why do the authors omit this data in the discussion? How will these differences be explained?
Reply: As suggested, we have added one paragraph in the Discussion section to describe this monthly variation of A. platys infection (Lines 226-235).
References - In the authors' names, diacritical marks appropriate to their languages ​​have been omitted. The correct spelling of names must be used. Example – 5, 7, 18…
Reply: As suggested, we have corrected the spelling with diacritical marks (Lines 276, 303).
Detail notices:
18 “all Taiwan strains” - the text shows the existence of one strain, why do the authors use the plural here? If they believe there is more than one strain, this should be justified.
Reply: As suggested, we have modified the word (strains -> specimens) for clarity (Line 18).
29 “a blue intraplatelet organism appeared in a stained blood film” -not in stained film, but in plates.
Reply: As suggested, we have modified the word as “blood smear” for clarity (Line 29).
38-39 “ dominant ectoparasite infesting canines around the world” – it is not true. R. sanguineus is the dominant parasite in the warm climate zone, limits of its distribution range are thought to be from latitude 20°N to below 30°S. In northern parts of Eurasia and Northern America it is absent.
Reply: As suggested, we have modified the sentence for clarity (Line 39).
87 “10X” – what does it mean? Is it 10×? The X (0078 Unicode, or 120 ASCII) sign means in the scientific record the unknown value. I suggest to replace it with the multiplication sign × (00D7 Unicode, or 215 ASCII); in computer text editors, it is no problem.
Reply: As suggested, we have replaced it with the multiplication sign (x) (Line 90).
196 “A. platys was first described” - this claim is not allowed in this place of the text! The authors of the manuscript were not the first to describe A. platys! This is the first identify of Anaplasma infection in R. sanguineus ticks infestig stray dogs of Taiwan.
Reply: We have modified the sentence for clarity (Line 197).
Round 2
Reviewer 1 Report
Comments and Suggestions for Authors
I agree with the revised version of the manuscript. The authors have addressed all the points raised.
Reviewer 3 Report
Comments and Suggestions for Authors
The present work can be accepted since all suggestions have been added to the final version of the manuscript.